# Chitin Derived Small Molecule AVR-48 Reprograms the Resting Macrophages to an Intermediate Phenotype and Decrease *Pseudomonas aeruginosa* Mouse Lung Infection

**Sumita Behera** [1,†]**, Santosh K. Panda** [2,†]**, Michael Donkor** [3]**, Eesha Acharya** [1]**, Harlan Jones** [3] **and Suchismita Acharya** [1,4,*]

1    AyuVis Research Inc., Fort Worth, TX 76107, USA
2    School of Medicine, Washington University, St. Louis, MO 63110, USA
3    Department of Microbiology, Immunology and Genetics, University of North Texas Health Science Center, Fort Worth, TX 76107, USA
4    Department of Pharmacology & Neuroscience, University of North Texas Health Science Center, Fort Worth, TX 76107, USA
*    Correspondence: sacharya@ayuvis.com
†    These authors contributed equally to this work.

**Abstract:** AVR-48 is a structural derivative of chitin previously shown by our laboratory to significantly decrease lung injury parameters in LPS, hyperoxia and sepsis-induced rodent models. The current study objectives are to determine the cellular mechanism of action and demonstrate efficacy in a mouse bacterial lung infection model. For in vitro receptor binding and macrophage polarization studies, C57Bl/6J mouse derived spleens and human peripheral blood mononuclear cells (hPBMCs) were treated with AVR-48 ± LPS or biotin conjugated AVR-48. Different macrophage types were determined using flow cytometry and secreted cytokines were measured using ELISA. In vivo, a CD-1 mouse *Pseudomonas aeruginosa* lung infection was treated with AVR-48, assessing bacterial colony forming unit (CFU), IL-10 and IL-17A levels in lung and blood samples. AVR-48 binds to both the toll-like receptor 4 (TLR4) and the CD163 receptor on mouse monocytes. In hPBMCs, frequency of intermediate macrophages increased upon AVR-48 treatment for 72 h. Increased bacterial phagocytosis/intracellular killing were observed in THP-1 cells and reduction in CFU in CD-1 mouse lungs. Binding of AVR-48 to both TLR4 and CD163 receptors bring the macrophages to an intermediary stage, resulting in increased phagocytosis and decreased inflammation, altogether providing an optimal immune balance for treating lung injury and infection.

**Keywords:** chitin analog; immunomodulation; TLR4; CD163; lung injury; bacterial infection; phagocytosis; anti-inflammatory; intermediate macrophages

## 1. Introduction

Chitin and chitosan are high molecular weight oligosaccharides with diverse biological activities, having anti-microbial, anti-inflammatory, anti-oxidant, anti-tumorigenic, immunostimulating, and calcium- and iron-absorption-acceleration properties [1]. Chitohexaose, a hexaoligomeric chitosan, binds to the Toll-Like Receptor (TLR4). TLR4 is a pattern recognition receptor (PRR) through which bacterial lipopolysaccharide (LPS) induces an innate immune response. Our previous studies suggested that [2] chitohexaose, after binding with TLR4, modulates macrophages to a non-inflammatory M2 phenotype and downregulates LPS induced inflammatory mediators such as IL-1β, TNF-α and IL-6. This mechanism of action (MoA) for chitohexaose was confirmed in a TLR4 mutant C3H/Hej mice model by Panda et al. [2]. The proprietary first generation novel chitohexaose analog, AVR-25, demonstrated marked efficacy in decreasing lung injury and inflammation in a hyperoxia-induced acute lung injury model in adult mice [3]. In addition, we demonstrated

that AVR-25 prevents lung injury in an hyperoxia-induced bronchopulmonary dysplasia (BPD) murine pup model [3,4]. Furthermore, AVR-25 was very effective in protecting mice against cecal ligation and puncture (CLP) induced polymicrobial infection and sepsis [5]. To allow an efficient manufacturing and formulation process, we further optimized the chemical structure of AVR-25 using a medicinal chemistry structure activity study to obtain a series of lower molecular weight compounds [6]. Recently we have reported that one of the optimized compound, AVR-48, significantly decreases lung injury, pulmonary edema, permeability and inflammation in both hyperoxia and LPS-induced Acute Lung Injury (ALI) and CLP-induced sepsis in murine models [3]. AVR-48 also was very effective in preventing hyperoxia-induced lung injury and pulmonary hypertension in an experimental bronchopulmonary dysplasia (BPD) model in new-born murine pups [7].

In this study, we investigated the possible immunomodulating MoA of AVR-48 by addressing the following key issues: (1) does AVR-48 have selective binding affinity to surface receptors on monocytes and/or macrophages over other mononuclear immune cells including T cells and B cells, (2) does AVR-48 have binding affinity to any other surface receptors on monocytes and macrophages besides TLR4, (3) does binding of AVR-48 to TLR4 agonize or antagonize the downstream signaling cascade, (4) does AVR-48 treatment to human peripheral blood mononuclear cells (hPBMCs) promote differentiation into different sets of macrophages (M1, M2 or an intermediate ($M_{int}$) phenotype) and whether these macrophages can phagocytose pathogenic bacteria, (5) does AVR-48 treatment reduce significant bacterial loads and inflammation in lung tissue following infection?

To address these questions, we used both murine splenic monocytes (MSMs) and hPBMCs for determining the binding efficacy of AVR-48 in mononuclear cell receptors, macrophage polarization and cytokine release studies. We used THP-1 human monocyte cell lines to test AVR-48 induced bacterial membrane phagocytosis and intracellular killing potential. Lastly, we demonstrated the in vivo antibacterial and anti-inflammatory activity using a mouse model of *Pseudomonas aeruginosa* lung infection.

## 2. Materials and Methods

### 2.1. Animals

Both female CD-1 and male C57BL/6J mice were purchased from Taconic (Rensselaer, NY 12144, USA) and were maintained in a breeding colony at the University of North Texas Health Science Center (Fort Worth, TX, USA) Animal Care Facilities with Animal Welfare Assurance No. D16-00417 (A3711-01). Animal procedures were performed in accordance with the NIH Guidelines for the Care and Use of Laboratory Animals and were approved by the Institutional Animal Care and Use Committee (IACUC) of UNTHSC, TX (Protocol no. 2020-0002).

### 2.2. Cells and Cell Lines

hPBMCs were directly purchased from Sigma-Aldrich, Inc. (St. Louis, MO, USA, catalog no. 690PB-100A) and were used as per manufacturer guidelines. THP-1 human leukemic monocyte cells were purchased from ATCC (American Type Culture Collection, Manassas, VA, USA, catalog no. TIB-202) and authenticated prior to receipt for their identity and validity. For THP-1 cells from all sources, the original stock was expanded and aliquoted into individual tubes prior to freezing in liquid nitrogen. The purpose was to avoid contamination of the stock and to maintain access to as much of the validated original sample as possible. Limiting the use of the products of each aliquot to 2–3 passages ensured maintenance of the integrity of the cell lines being used.

### 2.3. Bacteria

*Pseudomonas aeruginosa* (ATCC-BAA-2108) was purchased from ATCC and cultured for 16 h to obtain mid-log phase cultures on Brain-Heart Infusion Broth (BHI) [EMD, EMD Chemicals Inc. Gibbwtown, NJ, USA] plates. Mice were inoculated intranasally with *Pseudomonas aeruginosa* [$1 \times 10^6$ cells] in a volume of 20 μL of Brain-Heart Infusion Broth

[EMD] after anesthesia. For phagocytosis and intracellular cytotoxicity study, the GFP tagged *P. aeruginosa* (ATCC-10145GFP™) was used. For combination minimum inhibitory concentration (MIC) determination study, *P. aeruginosa* (ATCC-10145), *Acinetobacter baumannii* (ATCC-19606) and methicillin resistant *Staphylococcus aureus* (MRSA) (ATCC-BAA 1760) were purchased from ATCC and cultured in BHI.

### 2.4. Chemicals and Reagents

The synthesis and structural characterization of AVR-48 and biotinylated AVR-48 (BTAVR-48) was conducted in the laboratory of AyuVis Research Inc., following the in-house procedures [6]. Endotoxin-free phosphate-buffered saline (PBS), phorbol myristyl acetate (PMA), meropenem, imipenem, ciprofloxacin and colistin were purchased from Sigma-Aldrich Inc., St. Louis, MO, USA. Ultrapure LPS derived from *Salmonella minnesota* was purchased from InvivoGen, San Diego, CA, USA (catalog no. tlrl-smlps). Formulation of AVR-48 for in vitro and in vivo studies was done in saline.

For the mouse studies, AVR-48 was reconstituted in 0.9% sterile normal saline to prepare a final dose concentration of 10 mg/kg as a colorless solution, filtered with 0.2 μM syringe filter and injected IP (30 μL). For cellular assays, a 10 mM stock solution of AVR-48 was prepared in saline and further diluted with culture media to attain desired concentrations.

### 2.5. Cell Viability Assay

hPBMCs ($1 \times 10^5$) were plated in a 96 well plate and treated with either AVR-48 (0.1, 1.0, 10, 100 and 1000 μM), or BTAVR-48 (1, 10 and 100 μM) and incubated for 48 h. The cells were treated with MTT and resolving reagents (Promega, Madison, WI, USA), incubated for 2 h and read at 450 nm absorbance following manufacturer's instruction. The percentage of viable cells was calculated from the change in absorbance of the AVR-48 treated cells as compared to untreated control cells. Studies were performed in triplicate.

### 2.5.1. Cytokine and CD163 Assay

hPBMC cells ($1 \times 10^5$) were seeded in 96 well plates and treated with either AVR-48 (1, 10, 100 and 1000 μM) or LPS (50 ng/mL) or in combination for 6 h, 24 h, 48 h or 72 h. Cell supernatants were collected and assessed for IL-10 (catalog no. ELH-IL-10-1, RayBiotech, Peachtree Corners, GA, USA), for TNF-α (catalog no. ELH-TNF-α-1, RayBiotech, Peachtree Corners, GA, USA), for IL-6 (catalog no. ELH-IL-6-1, RayBiotech, Peachtree Corners, GA, USA) and for CD163 (catalog no. ELH-CD163-1, RayBiotech, Peachtree Corners, GA, USA) using a BioTek synergy H1 microplate reader (Fisher Scientific, Toronto, ON, Canada). All experiments were performed using two or three technical replicates.

### 2.5.2. IL-10 and IL-17A Detection in Lung Homogenates and Serum

Upon sacrifice, mice lung tissues were harvested and homogenized in PBS and stored at −80 °C until analysis. Serum samples were collected from anti-coagulated whole blood and stored at −80 °C until analysis. The concentration of IL-10 and IL-17A in total lung homogenates and serum were determined by sandwich ELISA method. All procedures were performed as described by the manufacturer. IL-10 and IL-17A concentrations were determined according to standard curve generated by reference concentration of cytokine at a wavelength of 450 nm detected by colorimetric plate reader (Biotek Instruments Inc. Winooski, VT, USA). ELISA antibody set and recombinant cytokine for standard were purchased from eBiosicence (eBiosciences, San Diego, CA, USA).

### 2.6. Flow Cytometry Studies
### 2.6.1. Binding of AVR-48 to Splenic Monocytes/Macrophages

Primary splenocytes of C57BL/6J mice (*n* = 3–5/group) were treated with AVR-48 (10 μM and 100 μM) for 72 h. PMA (200 ng/mL) was used as a positive control. The cells were washed and stained for CD11b (1:200) (catalog no. 101226, BioLegend, San Diego, CA, USA), MHC-II (1:500) (catalog no. 107630, BioLegend, San Diego, CA, USA), 7AAD

(catalog no. 559925, BD Biosciences, San Jose, CA, USA), LY6G (1:100) (catalog no. 127614, BioLegend, San Diego, CA, USA), Ly6C (1:100) (catalog no. 1280323, BioLegend, San Diego, CA, USA) and F4/80 (1:200) (catalog no. 123128, BioLegend, San Diego, CA, USA) and were acquired by flow cytometry using FACS Canto-II. The data were analyzed using FlowJo™ v 10.6.2 Software.

### 2.6.2. Binding of Biotinylated Conjugated AVR-48 to Splenic Monocytes/Macrophages

Briefly, cells ($1 \times 10^6$) were incubated at 4 °C for 1 h followed by incubation with biotinylated AVR-48 (0.25 μM, 0.5μM, 25 μM and 250 μM) along with monocyte (Ly6C) markers. Then the cells were probed with appropriate fluorescence-coupled streptavidin for anti-mouse TLR4 (1:100) (catalog no. SA-15-21 BioLegend, San Diego, CA, USA), TLR2 (1:100) (catalog no. QA16A01, BioLegend, San Diego, CA, USA), CD163 (1:100) (catalog no. S150491, BioLegend, San Diego, CA, USA) and CD206 (1:100) (catalog no. C068C2, BioLegend, San Diego, CA, USA) antibodies and were analyzed by FACS. Dead cells were excluded during analysis by live/dead staining.

### 2.6.3. Quantification of Macrophages after AVR-48 Treatment to hPBMC Cells

In a 96 well plate, 50,000 hPBMCs/well was stimulated with or without AVR-48 (1 μM, 10 μM and 100 μM) for 72 h at 37 °C under $CO_2$. The plate was spun at 1500 rpm for 7 min; supernatant was removed and stored at −80 °C. The cell pellet was incubated with ice cold PBS for 30 min to detach the cells followed by spinning the plate at 1500 rpm for 7 min and gently tapping to remove the media. A 20 μL/well of anti-CD32 antibody (cat. # 303202, BioLegend, San Diego, CA, USA) at 1:100 dilution was added and was incubated for 20 min at 4 °C followed by spinning the plate at 1500 rpm for 7 min and gently tapping to remove the media. Then 30 μL of the antibody cocktail of CD14 (1:50) (catalog no. 655114, BD Bioscience, San Jose, CA, USA), CD16 (1:25) (catalog no. 980104 BioLegend, San Diego, CA, USA), CD206 (1:50) (catalog no. 551135, BD Bioscience, Jose, CA, USA), CD86 (1:50) (catalog no. 555658, BD Bioscience, Jose, CA, USA), CD163 (1:50) (catalog no.: 556018 BD Biosciences San Jose, CA, USA) and HLA-DR, 5.25 μL, (1:200) (catalog no. 307617, BioLegend, San Diego, CA, USA) was added to each well and incubated for 30 min at 4 °C followed by washing with 200 μL of FACS wash buffer. The plate was spun at 1500 rpm for 7 min. Media was then removed and treated with 150 μL of live/dead 7AAD staining solution (1:50) (catalog no. 00-6993-50, Invitrogen, Waltham, MA, USA) and analyzed by BF-LSRII (Hampton, NH 03842, USA) flow cytometer, and FlowJo™ Software was used to analyze the results.

### 2.7. Phagocytosis and Bacteria CFU Measurement Using THP-1 Cells

THP-1 monocyte cells ($1 \times 10^5$) were treated with either PMA (400 ng/mL or 0.64 μM) or 200 μM of AVR-48 for 72 h. GFP tagged *Pseudomonas aeruginosa* (treated 1:20 ratio of cells: bacteria) was added and incubated for 0.5 h, followed by treatment with gentamycin (100 μg/mL) to remove extracellular bacteria. The intracellular bacteria colony forming unit (CFU) was determined after plating in agar. The intracellular killing percentage was calculated using the formula [CFU (0.5 h) − CFU (1 h)]/CFU (0.5 h) × 100% as described before [8].

### 2.8. Combination MIC Assay

The individual $MIC_{90}$ of the antibiotics meropenem, ciprofloxacin, colistin and compound AVR-48 was determined first using a broth dilution assay for *Pseudomonas aeruginosa* (ATCC-10145), *Acinetobacter baumanni* (ATCC-19606) and methicillin resistant *Staphylococcus aureus* (ATCC-BAA-1760) bacteria. To determine the synergy and the combination MIC of AVR-48 and different standard of care (SoC) antibiotic, a checkerboard design was used as described earlier [9]. In a 96 well plate, AVR-48 was added horizontally diluted two-fold from left to right where antibiotics were added from top to bottom at their MIC, 1/2 of MIC and at 1/4 of MIC concentrations. Then 5 μL of bacteria was added to each well

and incubated for 6 h; the plate was read at 600 nM. The $MIC_{90}$ was calculated from the change absorbance. Fractional inhibitory concentrations (FICs) were calculated by use of the following formula: FIC (X + Y) = [MIC of compound X in combination with Y]/[MIC of X alone]. The fractional inhibitory index ($\sum$FIC) was calculated as FIC of compound X + FIC of compound Y to evaluate interaction profiles. $\sum$FICs of $\leq$0.5 designates synergistic activity, $\sum$FICs of $\geq$4.0 indicates antagonism, and values in between correspond to additive, as outlined in previous work on antibacterial combination studies [10,11].

### 2.9. Pseudomonas aeruginosa Mouse Lung Infection

Briefly, groups of female CD-1 (*n* = 3–4/group) were lightly anesthetized and subjected to an inoculum of $1 \times 10^6$ CFUs of *P. aeruginosa* (ATCC BAA-2108) in two 20 µL droplets over the nares. Following an 8-h incubation period, groups of mice were intravenously administered with AVR-48 (10 mg/kg/dose, q12 h) alone, meropenem alone (1.5 mg/kg, q12 h) or in combination for 3 days. The compound was formulated in saline and injected using IV tail vein injection. One of the remaining groups was sham-treated with vehicle when the AVR-48 was administered, and the other two remaining groups remained untreated. Mice were euthanized at 72 h after infection, and blood and lungs were harvested from each animal. Total CFUs were determined from each blood and lung homogenates and processed for cytokine by ELISA.

### 2.10. Quantification of Bacterial Load in Mouse Lung and Blood Samples

Bacterial load in lung and blood were evaluated after infection. Upon sacrifice, lung tissues were harvested and homogenized in PBS. Ten-fold serial dilution of lung homogenates were plated onto blood agar plates and incubated overnight at 37 °C with 5% $CO_2$. Colonies on plates were enumerated, and the results were expressed as $\log_{10}$ CFU/mL per organ. Similarly, heparinized blood samples were serial diluted at 1:10 concentrations and plated onto blood agar plates. Colonies on plates were enumerated after overnight incubation at 37 °C with 5% $CO_2$. The results were expressed as $\log_{10}$ CFU per mL of homogenized organ.

### 2.11. Statistical Analysis

Statistical analysis was performed using GraphPad Prism Version 8.0 (GraphPad Software, San Diego, CA, USA). For multi-experimental group analysis, data were subjected to one-way ANOVA (analysis of variance) followed by post hoc tests (Dunnett's multiple comparison test, Newman-Keuls and Bonfferoni) for group differences. Homogeneous data were analyzed using the ANOVA, and the significance of intergroup differences between the control and test item-treated groups was analyzed using Dunnett's test. Heterogeneous data was analyzed using Kruskal–Wallis test and the significance of intergroup differences between the control and test item-treated groups were assessed using a nonparametric Dunnett's test. All data are reported as ±SEM. A significance level of $p < 0.05$ at 95% confidence intervals was considered statistically significant for all the experiments reported in this study. All the experiments are repeated two or three times with 2–4 technical replicates, and the data were pooled together and averaged for statistical analysis.

## 3. Results

### 3.1. AVR-48 and Biotin Conjugated AVR-48 Do Not Exhibit Cytotoxicity to hPBMCs In Vitro

First, the effect of AVR-48 and biotin conjugated AVR-48 on hPBMCs in vitro and cell viability was analyzed. Both of the molecules have no cytotoxicity to hPBMCs upon treatment for 48 h. The frequency of live cells was comparable between the control, AVR-48 (Figure 1A,C) or biotin conjugated AVR-48 (BTAVR-48) treatment groups (Figure 1B,D) when incubated using low-to-high concentrations. We have observed that upon AVR-48 (0.1 µM, 1 µM and 10 µM) treatment, there is increased cell proliferation as compared to control cells. The plausible reason might be differentiation of monocytes into macrophages

at 48 h which have a survival benefit to expand/proliferate as compared to the monocytes. Further investigation is warranted to decipher the exact mechanism.

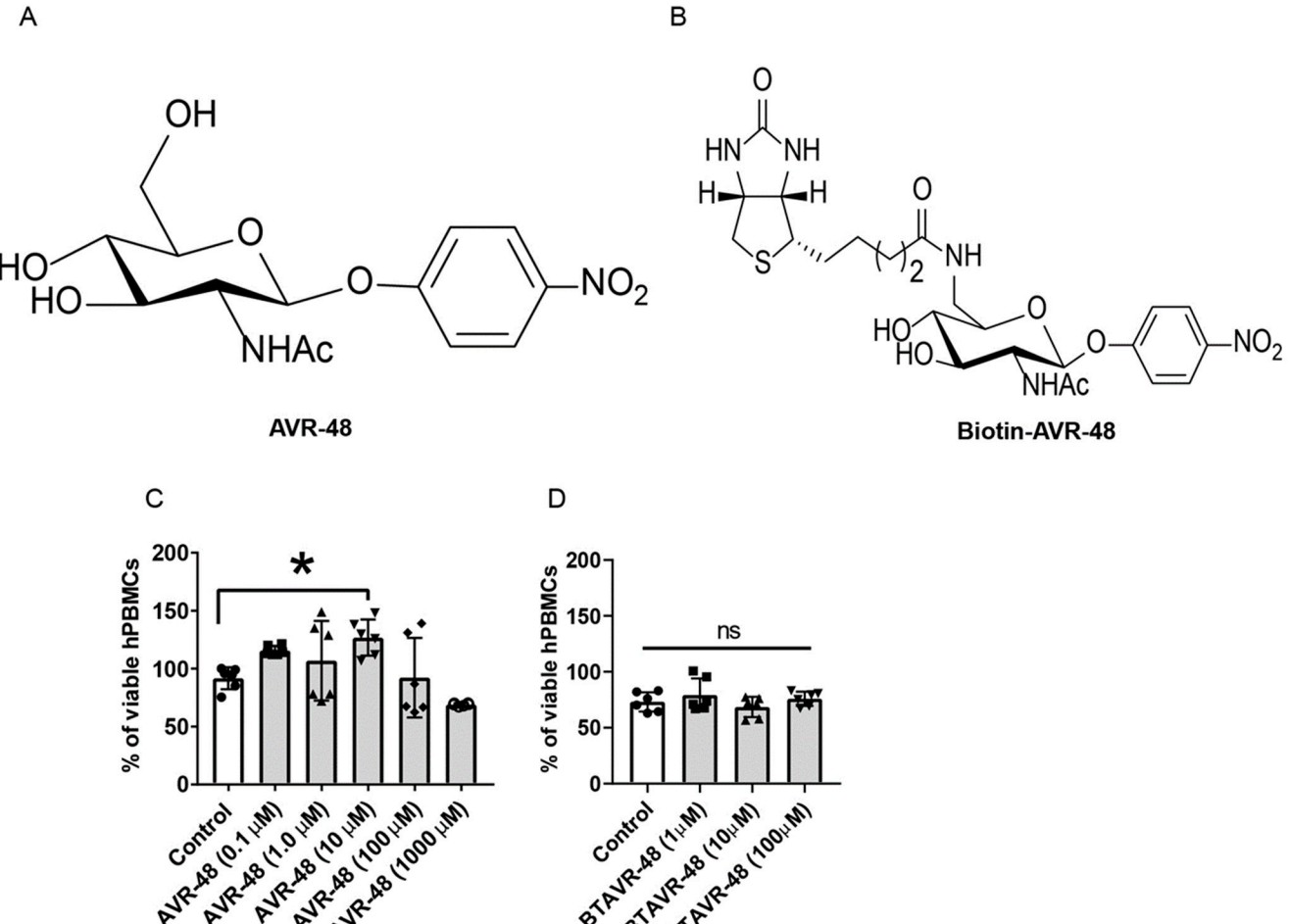

**Figure 1.** Assessment of cell viability after AVR-48 treatment to hPBMCs. (**A**,**B**) Chemical structure of AVR-48 and biotin-conjugated AVR-48 (BTAVR-48). (**C**,**D**) hPBMCs ($1 \times 10^5$) were plated in a 96 well plates and treated with either AVR-48 (0.1, 1.0, 10, 100 and 1000 μM) or BTAVR-48 (1, 10 and 100 μM) and incubated for 48 h. The cells were washed and treated with MTT reagents, incubated for 2 h and read at 450 nm absorbance. The percentage of viable cells was calculated from the change in absorbance of the samples as compared to untreated control cells. N = 2–3 technical replicates, experiments are repeated three times, and data presented are the averages of three experiments ± SEM. * $p < 0.05$, One-way ANOVA Graph Pad Prism v8.0.

### 3.2. AVR-48 Binds to Both TLR4 and CD163 Receptors in Primary Monocytes

In order to identify the on-target receptors for AVR-48 in the murine splenocytes, we performed a competitive binding experiment using antibodies to putative target and AVR 48. Binding of anti-TLR4 and anti-CD163 antibody to the surface of murine splenic cells was inhibited by BTAVR-48 in a dose-dependent manner (Figure 2B–E). The maximum binding efficacy (75–78%) was observed at 100 μM concentration for both receptors, and the $IC_{50}$ for BTAVR-48 was calculated to be 21.6 μM for TLR4 and 11.6 μM for the CD163 receptor from the dose response curve. We did not observe any binding affinity of AVR-48 for either TLR2 or CD206 receptors (Supplemental Figure S1).

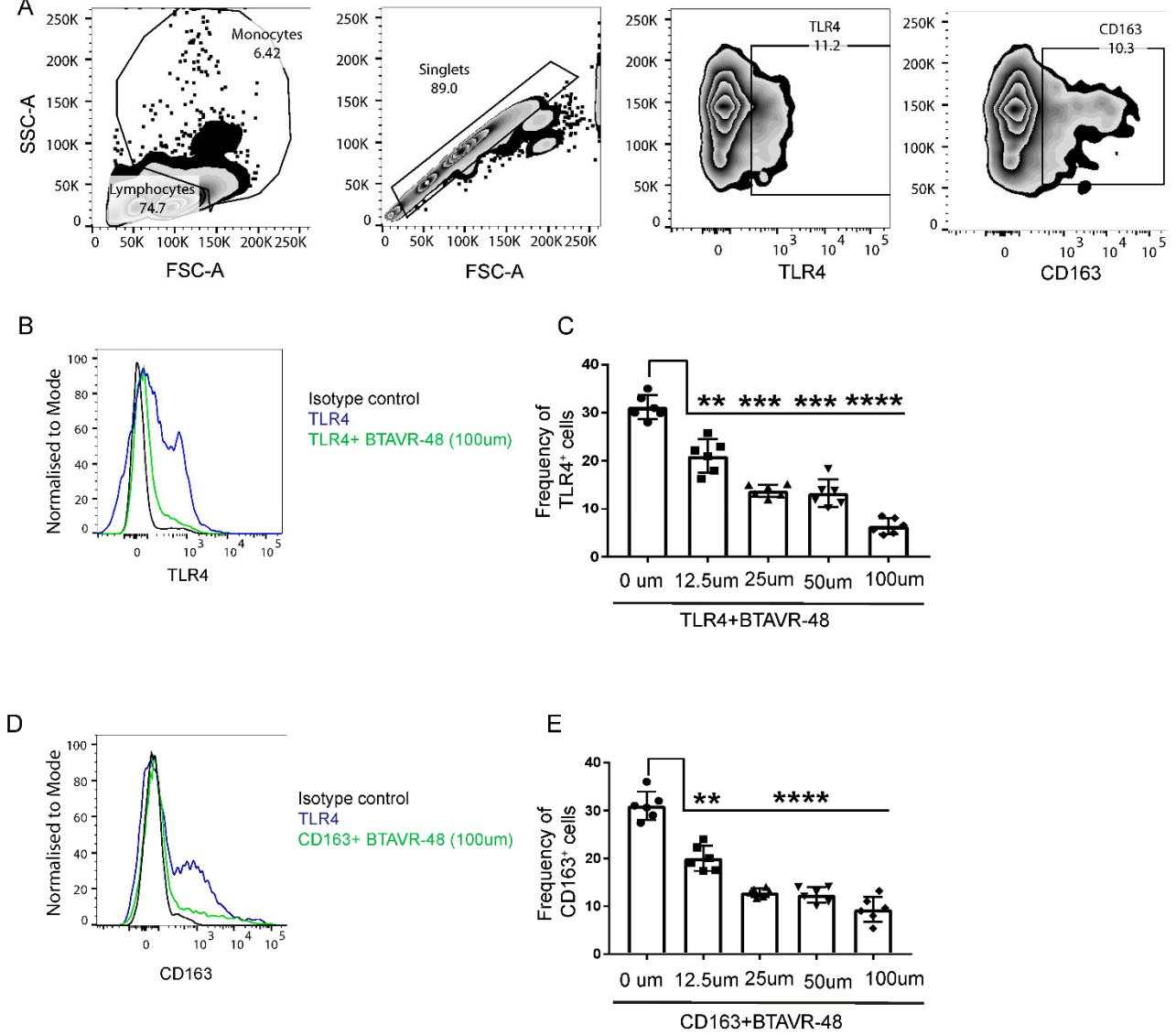

**Figure 2.** Binding study of biotin conjugated AVR-48 to mouse monocytes. Primary splenocytes from C57BL/6J mice (*n* = 3) were incubated at 4 °C for 1 h followed by incubation with biotinylated AVR-48 (BTAVR-48) at different concentrations along with monocyte (Ly6C) marker. The cells were probed with appropriate fluorescence coupled streptavidin (for anti-mouse TLR4 and CD163 antibodies), and percentage of cells is analyzed by Flow cytometry. (**A**) Gating strategy. (**B,D**) representative histogram plot depicting the inhibition of anti-TLR4 and anti-CD163 binding, (**C,E**) dose dependent inhibition of anti-TLR4 and anti-CD163 binding to the surface of monocytes. Each dot in the bar represents an individual mouse. Data were pooled from two individual experiments $\pm$ SEM. ** $p < 0.01$, *** $p < 0.001$, **** $p < 0.0001$, One Way ANOVA, analyzed by Graph Pad Prism.

### 3.3. AVR-48 Treatment Polarizes Mouse Monocytes to Macrophages and Shifts the Monocyte Populations More to a Resident Phenotype

Resident macrophages exert increased anti-inflammatory and phagocytic activities desired for resolution of inflammation, clearance of bacteria and tissue repair as compared to inflammatory macrophages [12]. In order to understand the mechanism behind our previously reported anti-sepsis and anti-inflammatory activity of AVR-48 in mouse models [3], we tested the effect of AVR-48 on macrophage polarization. Murine glass adherent splenic cells (considered as monocytes) [13] were stimulated with different doses of AVR-48 for 72 h and were analyzed for expression of different monocyte/macrophage markers. We observed that the expression of the macrophage-specific markers MHC-II and CD11b

(Figure 3B,C) were increased after AVR-48 treatment in a dose-dependent manner. Further staining with macrophage-specific marker F4/80 demonstrated an increase in macrophage population (F4/80$^+$) after treatment with either 10 µM or 100 µM concentrations of AVR-48 as shown in Figure 3D.

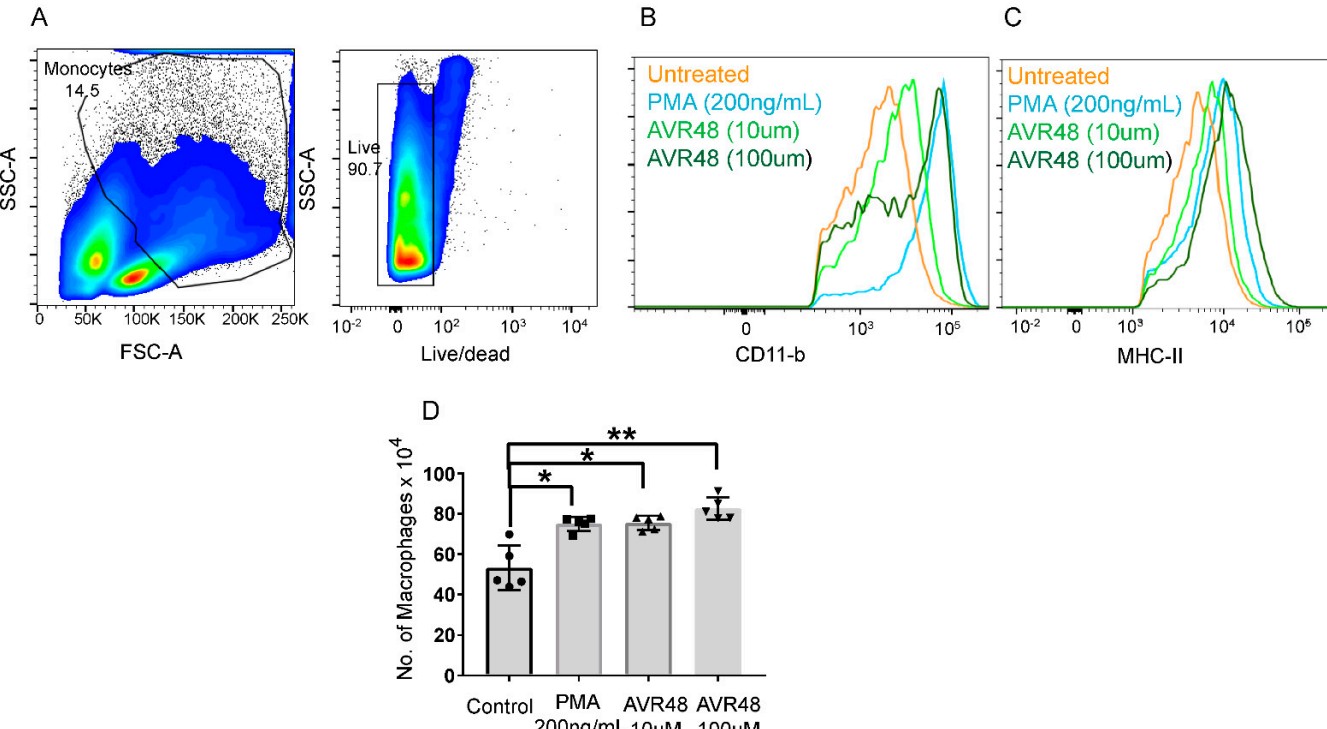

**Figure 3.** AVR-48 treatment and macrophage production. Primary splenocytes isolated from C57BL/6J mice (*n* = 3/group) treated with different doses of AVR 48 (10 µM and 100 µM) for 72 h. PMA (200 ng/mL = 0.32 µM) were used as a positive control. (**A**) Gating of lymphocytes and live/dead cells. The cells were washed and stained for CD11b and MHC-II, two surface markers for monocytes/macrophages, and analyzed by FACS. Live singlet cells, CD11b hi. CD11c low or—cells were gated further for F4/80 and MHC-II expression. Cells Hi for F4/80 and MHC-II were considered to be macrophages whereas cells low for F4/80 were considered to be monocytes. (**B**) AVR-48 treatment increased the MHCII positive cells in a dose-dependent manner. (**C**) AVR-48 treatment increased the CD11b positive cells in a dose-dependent manner. (**D**) Significant increase in the F4/80 positive cells representing macrophage population is observed at both 10 µM and 100 µM of AVR-48 treatment. Each dot represents an individual mouse. Data shown are a pool of two individual experiments ± SEM, * *p* <0.05, ** *p* < 0.001, One-way-ANOVA.

As described by Swirski et al. [13], in mice Ly6C$^{hi}$ cells are considered inflammatory monocytes whereas Ly6C$^{low}$ cells are resident and anti-inflammatory monocytes. A reciprocal increase of the resident cell population after treatment of splenocytes with AVR-48 for 72 h was observed (Figure 4B). Additionally, we confirmed that AVR-48's binding is specific to monocytes/macrophages. Binding of BTAVR-48 to splenic monocytes/macrophages showed a dose-dependent binding to monocytes/macrophages as determined by increased expression of MHCII+ and Ly6C$^+$ cells where there was no binding to T cells or B cells (CD3$^+$, CD19$^+$) observed with AVR-48 (Supplemental Figure S2A–C).

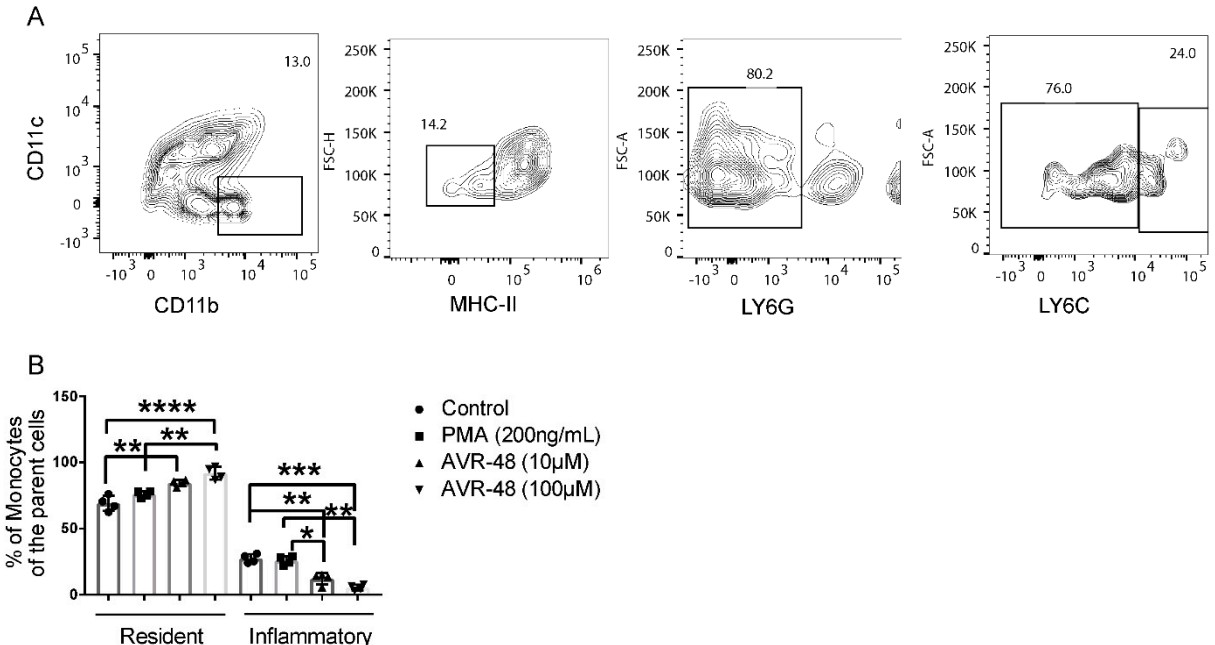

**Figure 4.** AVR-48 treatment increased resident monocytes after 72 h. (**A**) Primary splenocytes of C57BL/6J mice (*n* = 3/group) were treated of AVR 48 for 72 h. PMA (200 ng/mL) was used as a positive control. The cells were washed and stained for CD11b and MHC-II and were analyzed by FACS. Dead cells were excluded by live/dead staining during analysis. CD11b$^+$, MHC-II low/neg splenic cells with Ly6C hi are considered inflammatory whereas Ly6C low are considered resident and anti-inflammatory. (**B**) Treatment with both 10 μM and 100 μM concentrations of AVR-48 significantly increased the resident monocytes and decreased the inflammatory monocyte populations. The experiment is repeated two times. * $p < 0.05$, ** $p < 0.01$, *** $p < 0.001$, **** $p < 0.0001$. Two-way ANOVA, Tukey's multiple comparison test. GraphPad Prism 8.0.

### 3.4. Treatment of AVR-48 to Human Peripheral Blood Mononuclear Cells Increases the Percentage of Intermediate ($M_{int}$) Macrophages

hPBMCs were treated with different concentrations of AVR-48 (1 μM, 10 μM and 100 μM). FACS analysis after 72 h showed that AVR-48 treatment produced higher numbers of intermediate macrophages $M_{int}$ (CD14$^+$CD16$^+$HLA-DR$^+$CD206$^+$) compared to untreated control (Figure 5A,C). We did not observe significant number of $M_{int}$ at 48 h time point and observed a higher percentage of M1 macrophages (Supplemental Figure S4). The intermediate macrophages positive for HLA-DR are antigen presenting cells (APC) and anticipated to be more phagocytic and pro-inflammatory in nature to kill microbes [14,15]. Macrophages positive for CD206 are both phagocytic and anti-inflammatory in nature [16].

### 3.5. Effect of LPS and AVR-48 Treatment on Percentage of Macrophages in hPBMCs

In the combination study of AVR-48 with LPS in hPBMCs, we observed that the $M_{int}$ population (HLA-DR$^+$CD206$^+$) is significantly higher in both LPS-treated groups and LPS+AVR-48 treated groups (Figure 6B). No significant changes were observed with the M2 (HLA-DR$^-$CD206$^+$) macrophage populations between all groups. The overall percentage of HLA-DR$^+$CD206$^-$ macrophages noted here as M1 macrophages was very small (0.5–1.5% of the parent cells), and we observed no significant change in their numbers (Figure 6B) after LPS treatment. Interestingly, LPS+AVR-48 treatment groups at all concentrations showed a decrease in percentage of M1 macrophages (20–30%, $p < 0.05$) compared to either control or LPS-treated cells.

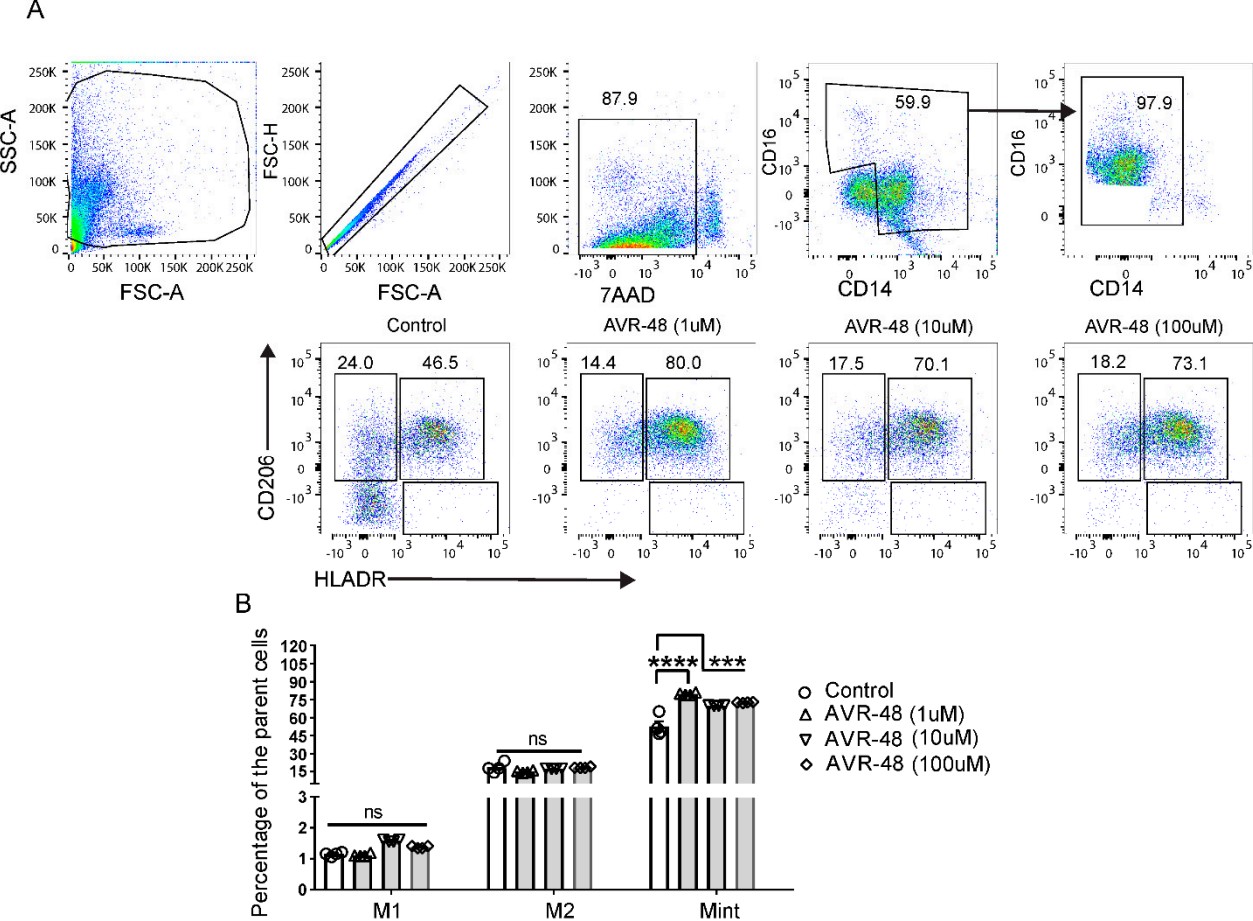

**Figure 5.** Change in macrophage populations after AVR-48 treatment. (**A**) Gating strategy and representative HLA-DR and CD206 staining of macrophages. Briefly, hPBMCs were plated in a 96 well plate and treated with AVR 48 for 72 h. The cells were washed and stained for CD32, CD14, CD16, HLA-DR, CD86, and CD206 anti-human antibodies and were analyzed by FACS. Dead cells were excluded by live/dead staining (7AAD) during analysis. The percentage of intermediate macrophages of the parent cells are determined as the macrophages stained positive for both HLA-DR and CD206 surface markers. (**B**) Represent the percentage of M1, M2 and $M_{int}$ macrophages of the parent macrophage populations (CD14$^+$CD16$^{++}$) after treatment with AVR-48. N = 2 technical replicates, the experiment is repeated four times, and data presented are the averages of the four experiments $\pm$ SEM. *** $p < 0.001$, **** $p < 0.0001$, One way ANOVA.

### 3.6. Effect of AVR-48 and LPS on Concentration of TNF-α and IL-6 in hPBMCs

Here we measured the concentrations of inflammatory cytokines TNF-α and IL-6 from the cell supernatant after treatment with either AVR-48 or LPS or both at four different timepoints. There were no significant TNF-α or IL-6 concentrations detected at 6 h and 24 h timepoints after treatment with AVR-48 compared to untreated control cells. There was a significantly ($p < 0.05$) higher (ca. 1.5-fold) concentration of TNF-α found after treatment with 48 h at 1 μM concentration (Figure 7A) compared to untreated control cells where no significant increase was noticed at either 10 μM or 100 μM. Similarly, no significant change in IL-6 was noticed in 48 h (Figure 7C). In control cells, at 72 h, very low levels of TNF-α were detected in cell supernatant where IL-6 concentrations were significantly higher by comparison. To our expectation, AVR-48 at all concentrations significantly decreased IL-6 concentrations (Figure 7C).

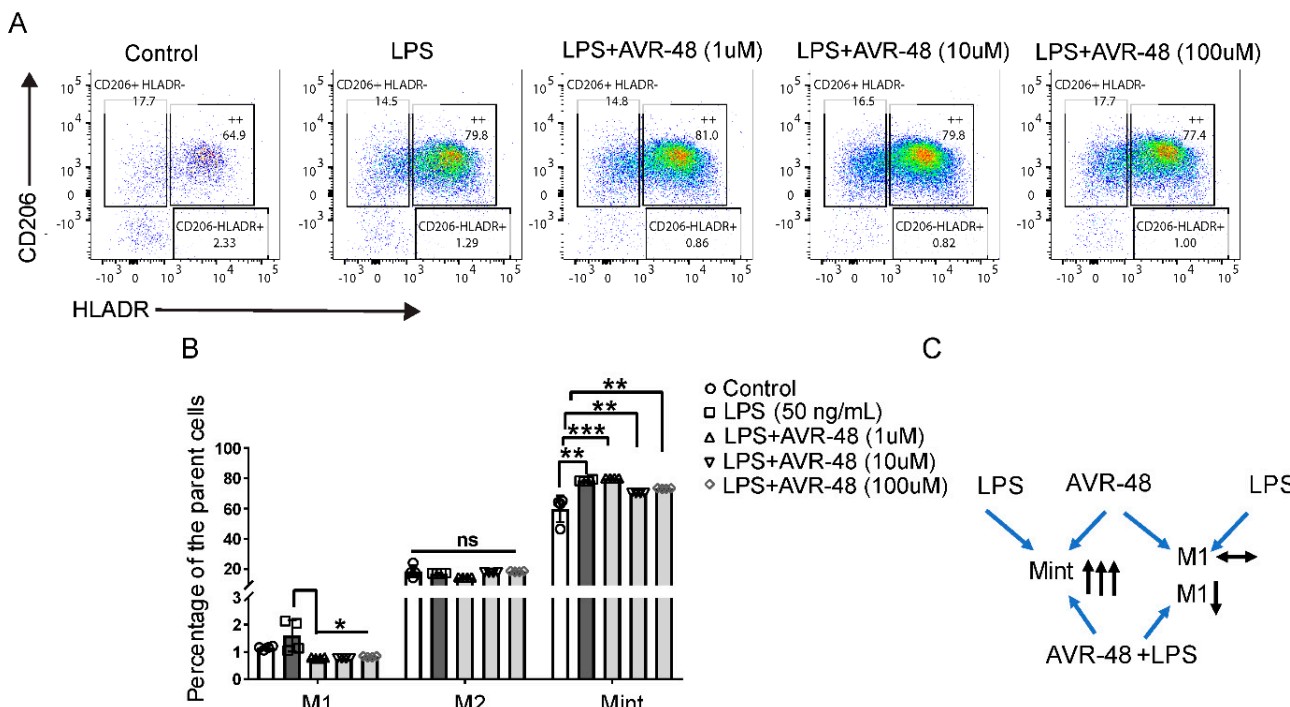

**Figure 6.** Change in macrophages populations after AVR-48 + LPS treatment. hPBMCs were plated in a 96 well plate and treated with LPS (50 ng/mL) or a combination with different concentrations of AVR-48 (1, 10 and 100 μM) for 72 h. The cells were washed and stained for CD32, CD14, CD16, HLA-DR, CD86, and CD206 anti-human antibodies and were analyzed by FACS. Dead cells were excluded by live/dead staining (7AAD) during analysis. (**A**) Representative FACS plot depicting the frequency of different macrophage populations. The percentage of intermediate macrophages of the parent cells are determined as the macrophages stained positive for both HLA-DR and CD206 surface markers. (**B**) Represent the percentage of M1, M2 and $M_{int}$ macrophages of the parent macrophage populations (CD14$^+$CD16$^{++}$) after treatment with AVR-48 + LPS. (**C**) Proposed mechanism of AVR-48. N = 2 technical replicates, and data presented as the averages of the four independent experiments ± SEM. * $p < 0.05$, ** $p < 0.01$, *** $p < 0.001$, One-way ANOVA.

LPS treatment significantly increased TNF-α at 6 h, 24 h and 48 h (Figure 7B). While LPS+AVR-48 treatment decreased the TNF-α concentration at a higher dose of AVR-48 (100 μM) after 6 h, AVR-48 was more potent at 24 h at all concentrations (1, 10 and 100 μM). No significant change in TNF-α concentration is observed at 48 h and 72 h with LPS + AVR-48 treatment. With LPS treatment, IL-6 concentrations were found to be significantly higher at early time points of 6 h and 24 h, but no significant change was noted at 48 h and 72 h (Figure 7D). Though we observed an increase in IL-6 after LPS + AVR-48 (100 μM) treatment at 24 h, after 72 h, IL-6 level was significantly lower compared to control cells (Figure 7D).

### 3.7. Effect of AVR-48 and LPS on Concentration of IL-10 and sCD163 in hPBMCs

In Figure 2 we showed that AVR-48 binds to the CD163 scavenger receptor in mouse primary spleen cells with an IC$_{50}$ of 11.6 μM. Here we show that in hPBMCs, 48 h treatment of AVR-48 (10–100 μM) significantly increases the level of soluble CD163 in the cell supernatant (Figure 8A). CD163 release is known to stimulate production of anti-inflammatory cytokine IL-10 which moderately increased (1.2–1.5-fold) at 48 h (Figure 8C). While LPS treatment increased the sCD163 (Figure 8A,B) and IL-10 concentrations at both 48 and 72 h (Figure 8C,D), a combination of LPS + AVR-48 significantly decreased the LPS-induced increase in both sCD163 and IL-10 concentrations.

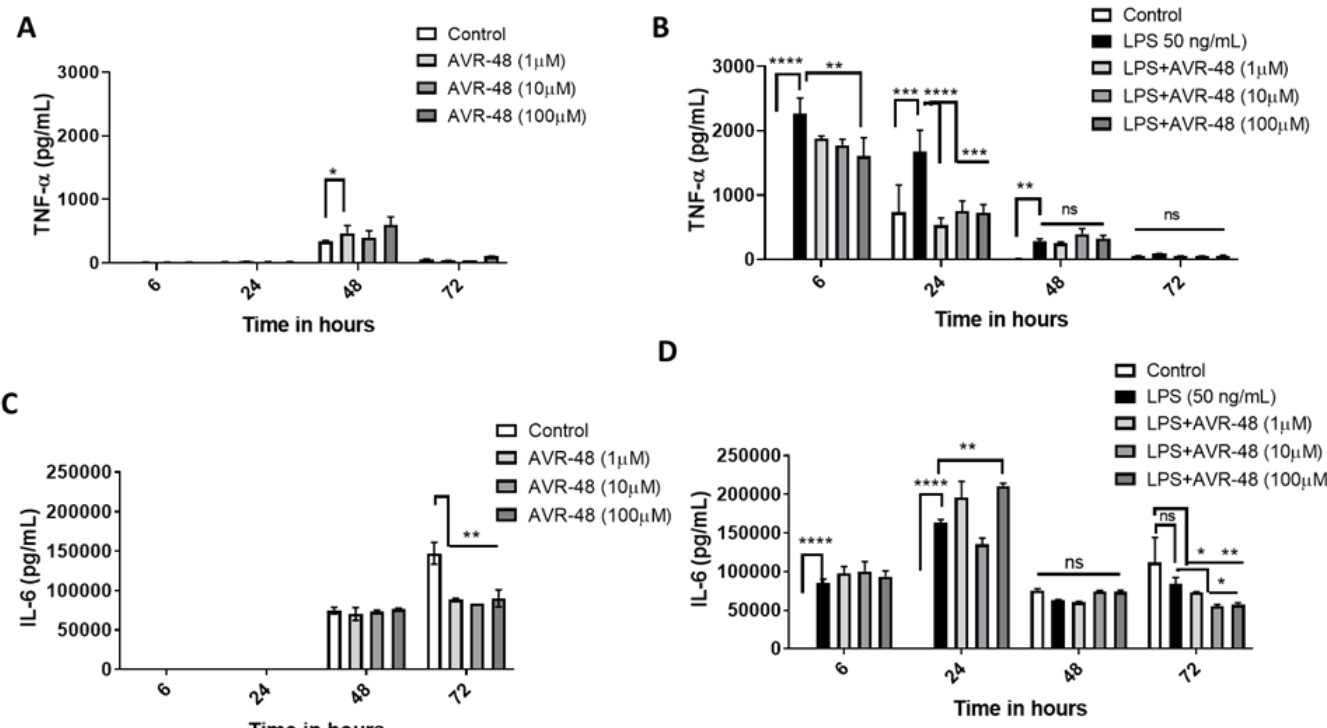

**Figure 7.** Assessment of cytokines after AVR-48 treatment at various time points. Briefly, hPBMCs were plated in separate 96 well plates and treated with AVR-48 (1, 10, 100 μM), LPS (50 ng/mL) or combination and incubated for 6 h, 24 h, 48 h or 72 h. The cell supernatant was analyzed for inflammatory cytokines TNF-α (**A**,**B**), and IL-6 (**C**,**D**) using ELISA. N = 3 technical replicates, the experiment is repeated three times, and data presented as the averages of three experiments ± SEM. * $p < 0.05$, ** $p < 0.01$, *** $p < 0.001$, **** $p < 0.0001$, Two-way ANOVA, Graph Pad Prism v8.0.

### 3.8. AVR-48 Induce Differentiation of THP-1 Human Monocytic Cells into Macrophages with Enhanced Phagocytosis of the Bacteria and Promote Intracellular Killing

In order to determine the phagocytic ability of the intermediate macrophages (M$_{int}$) produced after treatment with AVR-48, we next used THP-1 cells for this study [17,18]. We observed an increased number of macrophages (CD11b$^+$) after 72-h treatment of AVR-48 at 200 μM concentration (Figure 9A). PMA was used as positive control. Next, when we exposed these macrophages for 30 min to *Pseudomonas aeruginosa* bacteria tagged for green fluorescein protein (GFP), there was a significant increase in phagocytosis of the bacteria (Figure 9B) and subsequent intracellular-killing of them (Figure 9C) as determined by the CFUs measurement within 30–60 min.

### 3.9. AVR-48 Demonstrates Synergy with Standard of Care Antibiotics

We also tested the synergy and additive bactericidal activity of AVR-48 with several of the standard of care (SoC) antibiotics using a serial broth dilution minimum inhibitory concentration (MIC) determination assay following the checkerboard method [9]. The MIC$_{90}$ for AVR-48 was found to be >200 μg/mL for all bacteria. Surprisingly, when AVR-48 was combined with SoC broad-spectrum antibiotics—meropenem, colistin, ciprofloxacin—the MIC$_{90}$ measurements of the antibiotics were lowered by two- to tenfold (Table 1), enhancing the anti-bacterial activity. In particular, the MIC$_{90}$ of meropenem was lowered twofold for *P. aeruginosa,* MIC$_{90}$ of ciprofloxacin was lowered twofold for *A. baumannii* and MIC$_{50}$ of colistin was lowered more than tenfold for MRSA. Fractional inhibitory concentration ∑MIC was found to be <0.5 for the combinations which is a measure of synergy and additive between the two modalities (Supplemental Figure S6).

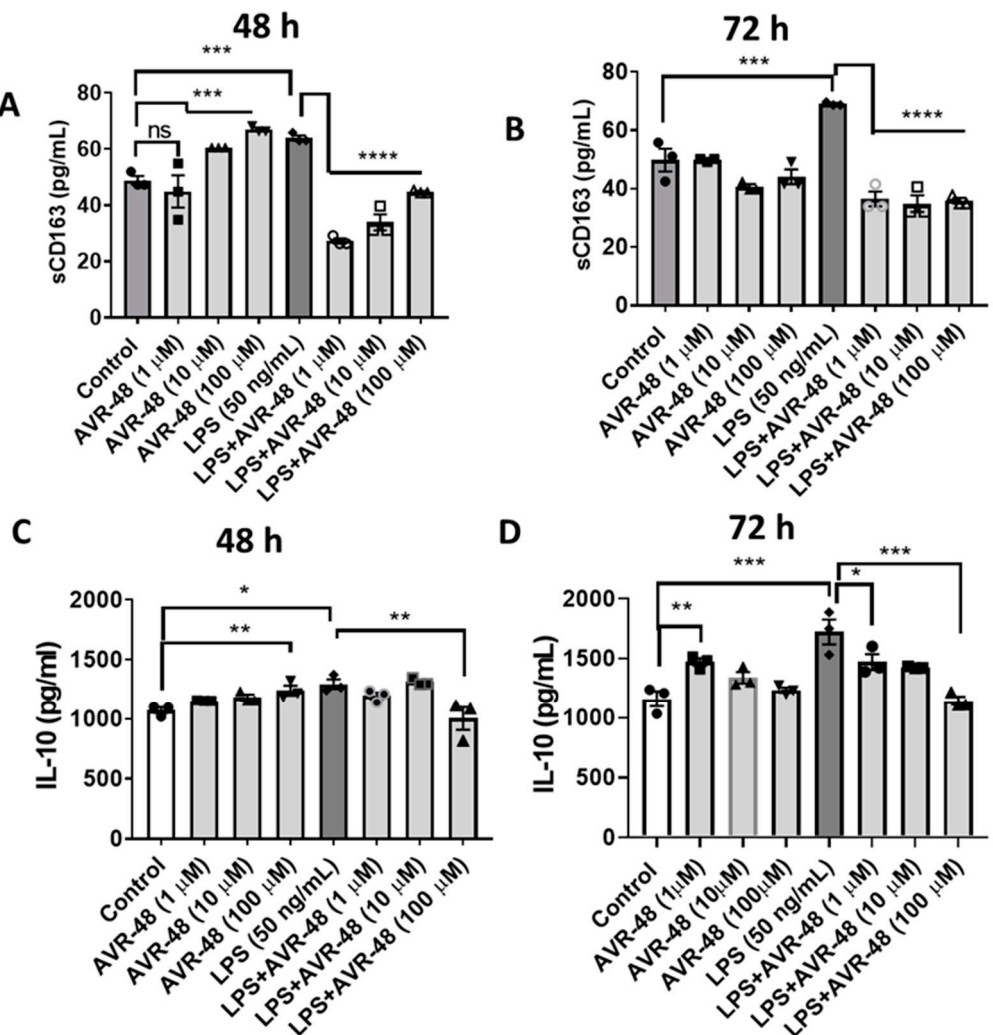

**Figure 8.** Assessment of IL-10 and sCD163 after AVR-48 treatment at various time points. Briefly, hPBMCs were plated in separate 96 well plates and treated with AVR-48 (1, 10, 100 μM), LPS (50 ng/mL) or combination and incubated for 48 h or 72 h. The cell supernatant was analyzed for IL-10 (**A**,**B**) and sCD163 (**C**,**D**) using ELISA. N = 2 technical replicates, the experiment is repeated three times, and data presented are averages of three experiments ± SEM. * $p < 0.05$, ** $p < 0.01$, *** $p < 0.001$, **** $p < 0.0001$, One-way ANOVA, Graph Pad Prism v8.0.

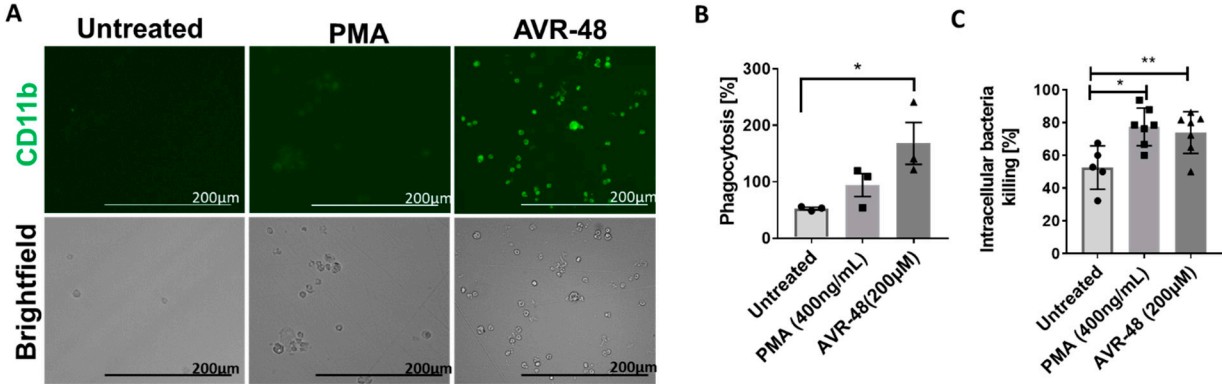

**Figure 9.** Phagocytosis and intracellular killing by AVR-48. (**A**) Monocyte to macrophage differentiation: THP-1 monocytic cells (1 × 10⁵) treated with PMA (400 ng/mL = 0.64 μM) or 200 μM of AVR-48

for 72 h produce macrophages (green, CD11b$^+$), scale bar = 200 μm. (**B**) Phagocytic activity of AVR-48: Increase in phagocytosis of GFP tagged *P. aeruginosa* (treated 1:20 ratio of cells: bacteria) was observed after 0.5 h of incubation and treatment with gentamycin (100 μg/mL) to remove extracellular bacteria. (**C**) The intracellular killing of *P. aeruginosa*: The intracellular bacteria CFU was determined after plating the cell lysate in agar. The intracellular killing percentage was calculated using the formula [CFU (0.5 h) − CFU (1 h)]/CFU (0.5 h) × 100%. N = 3–6 technical replicates and the data shown are averages of three independent experiments ± SEM. * $p < 0.05$ and ** $p < 0.01$. One-way ANOVA followed by Tukey post hoc analysis.

**Table 1.** Checkerboard combination MIC assay of AVR-48 + different antibiotic combinations using both gram-negative (*P. aeruginosa*, *A. baumannii*) and gram positive (MRSA) bacteria. To determine the synergy, AVR-48 was added horizontally, diluted twofold from left to right where antibiotics were added from top to bottom at their MIC, 1/2 MIC and 1/4 MIC concentrations in a 96 well plate. Then 5 μL of bacteria was added to each well and incubated for 6 h, and the plate was read at 600 nM. The MIC$_{90}$ was calculated. The columns 1–4 show individual MIC results, and the columns 5–6 in the table show the combination MIC doses for the respective antibiotics and AVR-48 (in brackets). N = 4 technical replicates, and the experiment is repeated thrice. The data presented as mean ± SEM. ND = not determined. * MIC$_{50}$ for MRSA.

| | MIC$_{90}$ (μg/mL) | | | | | | |
|---|---|---|---|---|---|---|---|
| **Bacteria** | **Mero** | **Cipro** | **Colistin** | **AVR-48** | **Meropenem (AVR-48)** | **Ciprofloxacin (AVR-48)** | **Colistin (AVR-48)** |
| *P. aeruginosa* (10145) | 4.0 | 2.0–4.0 | 8.0 | >200 | 1.5 ± 0.3 (4.6 ± 3.0) | 2.0 (3.6 ± 2.3) | 2.0 (3.6 ± 2.3) |
| *A. baumannii* (19606) | 0.5–1.0 | 2.0–4.0 | ND | >200 | ND | 1.0 (4.6 ± 3.0) | ND |
| *MRSA** (BAA 1760) | ND | ND | >200 | >200 | ND | ND | 14.5 ± 9.5 (4.6 ± 3.0) |

*3.10. AVR-48 Decreases Bacteria Load in a Mouse Lung Infection Model and Increases Anti-Inflammatory Cytokine IL-10*

Recently we published that AVR-48 at 10 mg/kg/dose significantly decreases lung injury, pulmonary edema, lung permeability and inflammatory cytokines in hyperoxia or LPS-induced acute lung injury (ALI) and cecal-ligation-and-puncture-induced (CLP) sepsis mouse models [3]. AVR-48 also was very effective in preventing hyperoxia-induced BPD and pulmonary hypertension in new-born murine pups [7]. Here we have demonstrated that AVR-48 (IV, 10 mg/kg, 2/d for 3 days)—alone and in combination with a standard of care antibiotic, meropenem—decreased bacterial CFUs in the lungs of mice infected with *P. aeruginosa* in a mouse model (Figure 10). Infection with *P. aeruginosa* increased the bacteria CFU significantly, and treatment with either meropenem or AVR-48 were equally effective in lowering the CFU level to roughly one log reduction. The combination AVR-48+meropenem group demonstrated a more robust (ca. 1.5-log-fold, ca. 95%) reduction in CFU, demonstrating excellent synergy and additive effect (Figure 10A). Also, the inflammatory cytokine IL-17A was found to be lowest in both lung and blood samples from AVR-48 and the AVR-48+meropenem group (Figure 10B) where the anti-inflammatory cytokine IL-10 concentration was significantly higher (Figure 10C).

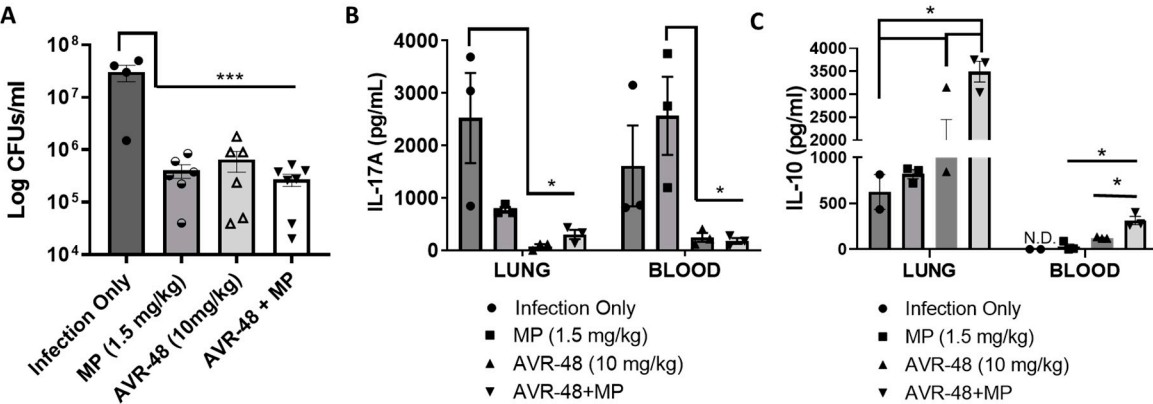

**Figure 10.** AVR-48 treatment in a lung infection mouse model. CD-1 mice (16 weeks of age) were housed in microisolator cages ($n = 4–5$) for two weeks prior to infection. Anesthetized mice were infected with $10^5$ CFUs of *P. aeruginosa* in 40μL by intranasal inoculation. Eight hours after infection, selected groups of mice ($n = 4–5$) were dosed with either meropenem (MP, 1.5 mg/kg), AVR-48 (10 mg/kg) or MP+AVR-48 in 200 μL saline IV (q12h) for 3 days. (**A**) Data represent the mean CFUs in lung at 72 h between treatment groups. Individual observations represent the number of mice remaining at time of sacrifice that were pooled from 2–3 replicate experiments. (**B**,**C**) Data show that, the concentrations of IL-17A and IL-10 from the serum or lung homogenate determined using ELISA after 72 h are lower than infection only groups. $n = 3$ for all groups. * $p < 0.05$, *** $p < 0.001$. One-way ANOVA.

## 4. Discussion

Low molecular weight chitin and chitosan inhibit sepsis caused by gram-negative bacteria via activation of an alternate immune pathway and possibly produce non-classical M2 macrophages with anti-inflammatory properties [6,9]. Specifically, we have shown previously that chitohexaose binds to and inhibits TLR4, the primary receptor to which LPS binds and mediates the consequent acute immune response [9]. But at the same time, activation of TLR4 signaling is also essential for survival in acute lung injury induced either by virulent *P. aeruginosa* type III secretory toxins found in multi-drug resistant (MDR) strains [5]. Thus, it is required to maintain a balance between inflammation and resolution in order to clear the bacterial load, decrease the hyper inflammation and minimize tissue injury. Key to its therapeutic efficacy, the binding of chitosan/chitohexaose to TLR4 happens in such a way that there is selective activation of the target cell and increase in expression of the important anti-inflammatory cytokine IL-10 [9]. The anti-inflammatory cytokine IL-10 is an important endogenous regulator of several pro-inflammatory cytokines (IL-1β, IL-6 and TNF-α) and chemokine expression in acute lung inflammation [10].

AVR-48 is an optimized second-generation structural analog and is a derivative of chitohexaose with improved physicochemical properties. Based on our published pharmacokinetic study [7], we demonstrated that the therapeutic activity of AVR-48 is systemic, and immune cells circulating in blood are the possible target cells for the mode of therapeutic action shown by AVR-48. Macrophages are tissue-resident professional phagocytes and antigen-presenting cells (APC), which differentiate from circulating monocytes. They perform important active and regulatory functions in innate as well as adaptive immunity [19]. Primary splenocytes serves as the major pool of lymphocytes and monocytes that further get activated to inflammatory (M1) or non-inflammatory (M2) macrophages depending on exposure to various physiological and pathological stimuli [20]. Macrophages are highly heterogenous cells that can rapidly change their function in response to local microenvironmental signals which can allow conversion of the initial M1/M2 polarization processes; for example, M2-polarized macrophages can convert to the M1-activated status under certain conditions and vice versa. The classically activated M1 macrophages comprise immune effector cells with an acute inflammatory phenotype. These are highly aggressive against bacteria and produce large amounts of lymphokines [21]. The alterna-

tively activated, anti-inflammatory M2 macrophages can be separated into at least three subgroups. These subtypes have various different functions, including regulation of immunity, maintenance of tolerance and tissue repair/wound healing. The general protocol of in vitro differentiation of monocytes into M1 or M2 macrophages takes a minimum of 6 days and the use of either granulocyte-macrophage colony-stimulating factor (GM-CSF) or M-CSF and a combination of cytokines/chemokine cocktails. To our utmost surprise, we found that treatment with 100 µM of AVR-48 for 3 days differentiated the primary mouse splenocytes into macrophages (ca. 80%). Swirski et al. [13] reported that mouse spleen cells with Ly6C$^{Hi}$ are considered inflammatory whereas Ly6C$^{low}$ are considered resident and anti-inflammatory. Here we found a higher percentage of resident monocyte populations after AVR-48 treatment and a lower number of inflammatory monocytes. Most importantly, we found that AVR-48 demonstrates selectivity for binding to monocytes and not to other mononuclear immune cells including T cells and B cells. Encouraged by this finding, we conducted further studies to identify what is (are) the putative target receptor(s) to which AVR-48 binds and to agonize/antagonize the receptor(s) to provide the therapeutic activities. We first probed for both TLR4 and TLR2 receptors. Using a biotin tagged AVR-48, we confirmed the binding of AVR-48 to TLR4 and selectivity over TLR2 in mouse spleen cells. What was important at this point was to delineate further to find out whether binding of AVR-48 to TLR4 protein expressed in the surface of monocytes/macrophages leads to activation, inhibition or modulation of the receptor protein. In our published work using a juvenile mouse model of BPD [7], we demonstrated that AVR-48 treatment to both room air (RA) control and hyperoxia (100% O$_2$) mouse pups increased the TLR4 protein concentration in lung tissues after 10 days post last dose of AVR-48 treatment. This increased TLR4 protein concentration coupled with an increase in lung cytokines TNF-$\alpha$, IL-6, IL-1$\beta$ and IL-10 supported possible activation of the TLR4 pathway by AVR-48 in naïve animals. However, the concentration of these inflammatory cytokines was significantly lower in both lung and plasma samples obtained from hyperoxia+AVR-48 mouse pups as compared to only hyperoxia groups, indicating a differential MoA by AVR-48 in physiological and pathological scenario which was intriguing.

Our macrophage differentiation study confirmed that there is a fluidic event of macrophage polarization happening after treatment of AVR-48 to hPBMCs. After 48-h treatment with either AVR-48 or AVR-48+LPS groups, the monocytes are predominantly differentiating into macrophages (80–90%) positive for HLA-DR antigen, M1 marker CD86 and negative for M2 marker CD206 (Supplementary Figure S4); at 72 h the percentage of intermediate (M$_{int}$) macrophages positive for HLA-DR antigen, M1 marker CD86 and M2 marker CD206, dominate (70–80%). The intermediate macrophages positive for HLA-DR are antigen-presenting cells (APC) and anticipated to be more phagocytic and pro-inflammatory in nature to kill microbes [14,15]. Low-level HLA-DR antigen expression on peripheral blood monocytes correlates with infection [14]. Macrophages positive for CD206 are both phagocytic and anti-inflammatory in nature [16]. The activation of monocyte or resting macrophages (M0) to M1 phenotype at an earlier timepoint can be explained as activation of TLR4 by AVR-48 similar to the known TLR4 agonist LPS. However, it was not very clear why there is an alternate activation and how the percentage of CD206 (M2 phenotype) macrophages are increasing at a later timepoint. In general, macrophage colony stimulating factor (M-CSF) and IL-10 are known to stimulate the macrophages to M2 phenotypes [22]. Here, we did observe an increase in IL-10 concentration in hPBMCs supernatants as well as in the plasma and lung from the mouse model of lung infection.

Additional investigation was made to identify any alternative on-target effect of AVR-48. We hypothesized that AVR-48 might have binding affinity to other receptors specific to M2 macrophages such as scavenger receptor CD163 or mannose receptor CD206 that are expressed abundantly in M2 macrophages. CD163 is a transmembrane scavenger receptor found on the surface of macrophages. Both CD163 and CD206 are expressed by macrophages, but the expression of the receptors is differentially regulated. CD163 expression is increased in response to IL-10 stimulation, while CD206 expression is upregulated

by IL-4 and IL-13 [23]. Procherary et al. [24] demonstrated that the activation state of CD163 and CD206 is reversible depending on the microenvironment, suggesting that a given cell may participate sequentially in both the induction and the resolution of inflammation. Cell surface expression of CD163 on alveolar macrophages is reduced in human subjects with asthma [25], BPD [26] and sepsis which suggests that CD163 may participate in the regulation of airway inflammatory responses in the lung. CD163 is released in the circulation in its soluble form, sCD163, via cleavage of the extracellular domain by matrix metalloproteases (MMPs) following oxidative stress [27] or via activation of TLR4 after inflammatory stimuli [28,29], and it is used as a biomarker for increased injury/inflammation. sCD163 protects monocytes from hyperactivation during bacterial infections by dampening the secretion of the proinflammatory cytokines TNF-α, IL-1β, IL-6 and IL-8 [30] and by endocytosing inflammatory neutrophils, excess hemoglobin/haptoglobin complexes responsible for increasing reactive oxygen species (ROS), as well as bacteria/viruses, possess phagocytic activity in clearing both microbes and cell debris. Here, we demonstrated that AVR-48 binds to the CD163 receptor in mouse splenocytes in a dose-dependent manner with an $IC_{50}$ of 11.6 μM. CD163 stimulation is known to produce anti-inflammatory cytokine IL-10. We did observe an increase in IL-10 concentration after 48 h of treatment with AVR-48 in hPBMCs which indicated that binding to CD163 receptor resulted in receptor stimulation. IL-10 is also known to negatively regulate production of IL-1β [19,31], an important endogenous regulator of chemokine expression in acute lung inflammation [20]; appears to be protective against the development of BPD [32–37], and IL-10-induced microRNA-187 negatively regulates TNF-α, IL-6 and IL-12p40 production in TLR4-stimulated monocytes [38]. The time and dose-dependent IL-10 and sCD163 measurement in plasma at 48 and 72 h after treatment with AVR-48 or LPS or both to PBMC showed that there is a dose-dependent increase in sCD163 and IL-10 at 48 h with AVR-48 treatment without significant change in TNF-α level. However, LPS-treated cells showed an increase in sCD163 and IL-10 along with increased concentration of TNF-α at the 6-h to 48-h timepoints. This further confirms that increased IL-10 concentration by AVR-48 is not exclusively via TLR4 activation, but the activation of CD163 contributed to this outcome as well. The decrease in sCD163 and TNF-α at 72 h after LPS+AVR-48 treatment demonstrated the consorted effort of CD163- and TLR4-mediated IL-10 production to negatively regulate TNF-α and IL-6, bringing all cytokine levels close to control creating a feedback regulation. The increase in IL-10 and decrease in inflammatory cytokine IL-17A in mouse lung and blood corroborated with the in vitro results.

LPS is a known TLR4 agonist and a secreted toxin of the gram-negative bacteria, including *P. aeruginosa* that activates TLR4 via the MyD88 pathway, activating the nuclear transcription factor NFkB and the release of inflammatory cytokines TNF-α, IL-1β, IL-6 and other chemokines. These molecules trigger other immune cells to act together as the first line of defense to kill invading bacteria and other microbial pathogens via increased phagocytosis and enhanced intracellular killing. So, we believe that activation of both TLR4 and CD163 by AVR-48 to differentiate the macrophages to an intermediary phenotype might be the cause of the key results: increased phagocytosis and decreased bacterial load in the mouse model of *P. aeruginosa* infection as well in the in vitro THP-1 phagocytosis assay. AVR-48 by itself did not demonstrate any direct bactericidal activity, and the concentration where 90% of bacteria get killed ($MIC_{90}$) was ≥200 μg/mL when tested for several gram positive (MRSA, MSSA) and gram negative (*P. aeruginosa*, *A. baumannii*, *E. coli*, *K. pneumonia*) using a broth dilution assay (Supplementary Figure S5). Surprisingly, when AVR-48 was combined with several of the SoC broad-spectrum antibiotics—meropenem, colistin, ciprofloxacin—the $MIC_{90}$ of the antibiotics were lowered more than four- to tenfold, enhancing the anti-microbial activity and supporting the concept of synergy/additive activity [13] between these two modalities with an unknown mechanism that needs further investigations.

In summary, using primary spleen derived mononuclear cells, we showed that a small molecule AVR-48 derived from chitin with N-acetyl glucosamine as a core structure has a

unique immunomodulating activity that restores the balance between activation and inhibition of innate immune response when exposed to LPS or to gram-negative bacteria. AVR-48 binds selectively to both TLR4 and CD163 receptors present in monocytes/macrophages with sub-micromolar $IC_{50}$ and has no binding affinity for T cells or B cells. The net effect of the binding of AVR-48 to both TLR4 and CD163 receptors was an increase in percentage of intermediate macrophages population ($M_{int}$) positive for both HLA-DR antigen and mannose receptor CD206, rendering both phagocytic and anti-inflammatory activities when tested in both in vitro and in vivo scenario using gram-negative bacteria *Pseudomonas aeruginosa*. In an in vivo model of a *P. aeruginosa*-induced lung infection in mice, AVR-48 treatment produced a significant decrease in lung bacteria counts; inflammatory cytokine IL-17A showed improvement of gross lung pathology. Increased concentrations of the anti-inflammatory cytokine IL-10 was observed in both mouse lung and plasma. Moreover, we observed a synergy and additive effect of AVR-48 with several SoC antibiotics providing an opportunity for both mono and combination therapy a during bacterial lung infection.

**Supplementary Materials:** The following supporting information can be downloaded at: https://www.mdpi.com/article/10.3390/immuno2040040/s1, Figure S1: Binding study of biotin conjugated AVR-48 to mouse monocytes; Figure S2: Binding of biotinylated conjugated AVR-48 to splenic monocytes, T cells and B cells; Figure S3: AVR-48 treatment increased resident monocytes in mouse spleen cells after 72 h; Figure S4: Flow cytometry data after treatment of AVR-48 to hPBMCs for 48 h; Figure S5: Determination combination minimum inhibitory concentrations (MIC90) of AVR-48 and different antibiotics for *Pseudomonas aeruginosa*.

**Author Contributions:** S.A., S.K.P. and H.J. conceptualized and designed the studies; M.D., S.B. and E.A. carried out the experiments; S.A., S.K.P. and H.J. analyzed the data; S.A., S.K.P. and H.J. prepared the initial drafts of the manuscript. All authors have read and agreed to the published version of the manuscript.

**Funding:** This work was partly funded by NIH in the form of SBIR grant 1R43AI129164-01 awarded to AyuVis Research.

**Institutional Review Board Statement:** Animal procedures were performed in accordance with the NIH Guide for the Care and Use of Laboratory Animals and were approved by the Institutional Animal Care and Use Committee (IACUC) of University of North Texas Health Science Center, FW, TX (Protocol # 2020-0002). No IRB protocol was required as the hPBMCs was purchased from commercial vendors.

**Informed Consent Statement:** Not applicable.

**Data Availability Statement:** Data is contained within the article or Supplementary Materials.

**Acknowledgments:** We are extremely thankful to Stella M Robertson, Suhas G Kallapur and Adam S Dayoub for critically reviewing the manuscript and providing their candid feedback.

**Conflicts of Interest:** The authors declare no conflict of interest.

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
