# Peer review of "Chitin Derived Small Molecule AVR-48 Reprograms the Resting Macrophages to an Intermediate Phenotype and Decrease Pseudomonas aeruginosa Mouse Lung Infection"

_2673-5601, doi:10.3390/immuno2040040_

Round 1

Reviewer 1 Report (Previous Reviewer 1)

I still have concerns regarding Fig 7.

Why cytokine secretion is drastically increased at 48h and 72h? At this time points, LPS doesn’t cause any additional difference. These data should be removed. Only effects on the LPS response at 6 or 24h have biological relevance.

In general, combining technical and biological replicates to achieve statistical significance is not legitimate, in my opinion. Technical replicates should be combined to generate one biological replicate and these should be assessed, with at least three independent biological replicates.

Author Response

Please find attached the response. Thanks

Reviewer 2 Report (Previous Reviewer 2)

The authors have re-submitted the result figures with the appropriate representative numbers missing in previous submissions and described the dot representation. The reviewer appreciates that the authors also expanded the concentrations and amounts of antibodies used.

The authors stated submitted that they would address 5 statements:

1. Does AVR48 selective binding affinity to phagocytes including lymphocytes- yes by monocytes, but not by T cells or B cells. Showed appropriate technical replicated and Lymphocytes are shown in supplemental figures

2. Does AVR48 get recognized by other TLR receptors- TLR4 and CD163 were noted to detect BTAVR48, TLR2 was in the supplemental figures with CD206

3. Does AVR48 agonize/antagonize downstream signaling- AVR48 demonstrated to be an agonist as effective as PMA at either concentration used 10 vs 100uM 

4. Does AVR48 skew hMDMs- no significance between M1 or M2 but noted  significant differences in M(intermediate), showed the appropriate amount of technical runs, noted proinflammatory mediators 

5. Does AVR48 treatment reduce bacterial loads and inflammation- Evaluated checkerboard MIC assay and Pseudomonas infection. and noted significant differences between infection only vs AVR48+Meropenem within 72 Hr. 

Minor:

Need clarification on the gating strategy of Figure 5. There's 2 gates describing CD16+CD14+ from live cells but it's unclear what the 59.9 vs 97.9 represent. Adding arrow to subgates may be helpful.

May need to include the role of IL-17a in the schematic illustration 

State sex and age of mice used 

Can you describe PMA concentration in uM to show comparisons between AVR48 vs PMA (200ng/ml = uM) used 

Author Response

Please find attached the response

Round 2

Reviewer 1 Report (Previous Reviewer 1)

The revisions made allow the publication of the manuscript in the present form.

This manuscript is a resubmission of an earlier submission. The following is a list of the peer review reports and author responses from that submission.

Round 1

Reviewer 1 Report

The article of Behera et al. describes the actions of the chitin derivative AVR-48 on inflammatory and anti-bacterial responses in different cell culture systems of murine and human origin (mouse splenocytes, human peripheral blood mononuclear cells (PBMCs) or monocytic THP-1 cell line). The authors show binding of AVR-48 to toll-like receptor 4 as well as scavenger receptor CD163 in splenic cells. They also show inhibitory effects of AVR-48 on LPS-stimulated TNFa secretion in PBMCs. Further, the authors report increased phagocytosis and intracellular killing of  Pseudomonas aeruginosa in THP-1 system as well as enhancement of bactericidal action of different antibiotics in a broth dilution assay. In in vivo setting, using pseudomonas aeruginosa mouse lung infection model, the authors observe reduced bacterial load in the lung as well as suppressed levels of pro-inflammatory cytokine IL-17A following AVR-48 treatment.

In my opinion, several flaws in data presentation and interpretation prevent the publication of the manuscript in its current form.

First, I doubt the validity of data presented in Fig 2D. Usually LPS induces much stronger responses of TNF, which is indeed the case for Figure 8A (what is the difference between these two experiments)  48 and 72h time points are also not optimal to study LPS-induced cytokine release, which occurs within several hours of LPS treatment.

Second, the very low percentage of CD206-negative, HLA-DR positive population of PBMCs referred to as M1 population makes difficult the interpretation of AVR-48 action. Particularly, LPS has no effect on this population, which questions then the relevance to the assignment of this population as pro-inflammatory.

Fig 6B-D and 7B-D only 2 data points are visible on the bar charts, although n=3 is stated.

Minor:

Graph abstract – iNOS is not a receptor.

Line 126 – In the Methods section, TLR4 analyses are described to be done in THP-1 cells. In the results, TLR4 levels in PBMCs are described. The relevance of measuring TLR4 levels is also not very clear.

Author Response

Please see attached the step by step response to all comments. Thanks

Reviewer 2 Report

Behera and colleagues characterized a novel chitohexaose analog  AVR-48 to evaluate immunomodulating mechanism of action and its efficacy in a Pseudomonas-murine pulmonary infection model. This is following previous studies AVR-48 evaluating in preventing hyperoxia-induced lung injury and pulmonary hypertension in experimental bronchopulmonary dysplasia model. 

The authors sought to address multiple questions including: 1. what is the binding affinity to receptors in macrophages and lymphocytes. 2. They also evaluated whether other receptors are involved in recognition of AVR-48 besides TLR4, 3 whether TLR4-AVR-48 is agonizing or antagonizing downstream signaling, 4 what is AVR-48's role in macrophage interaction and polarization, 5 would AVR-48 reduce lung bacterial load in a murine model and finally would it have synergistic effect in combination with antibiotic treatment.  

Overall the manuscript is well written and ambitious, as they set out several questions in this manuscript. They characterized a variety of mechanistic questions characterizing AVR-48. It was interesting to note that the authors did discover quick treatment of monocytes using AVR-48 allows for maturation into macrophages in less than 3 days and skews them towards M1 phenotype with M1 and M2 markers being most prevalent. 

The authors evaluated cell uptake by macrophages and brings back various cell types, including T cell and B cells, and it appears that monocytes, particularly low resident monocytes are brought in after 72 Hr. 

They did evaluate cells that took up AVR-48 and assisted in phagocytosis and killing of Pseudomonas by THP1 cells.

The authors also demonstrated that antibiotic treatment in addition to AVR 48 was labor to decrease CFUs while preventing the induction of IL-17, but did increase IL-10 levels. 

Major comments: Authors need to state the IRB protocol#, number of samples, technical #s used and source for hPBMCs.

Could the authors discuss the number of wells/donors used for the viability assay (Fig 1C). some conditions have 4 values while others are 3.  Additionally AVR-48 treatment appears to improve viability, I assume increase cell count at 1.0 and 10 uM, could the authors address this phenomenom?

It was my understanding that TAK242 was an antagonist to TLR4 however the cytokine levels of TNFa compared to LPS alone (+) didn't show differences. Perhaps is due to alternative pathways compensate for production of TNF, did the authors evaluate IL6 or other pro inflammatory mediator? LPS+AVR-48 did not have an additive effect, while AVR-48+TAK242 did decrease.

It is interesting to see that AVR-48 treatment induces increase MHCII, this would be interested to see in future manuscripts how mature the macrophages treated with AVR-48. 

Have the authors consider using AVR-48 treatment against fungal/parasitic pathogens? Exposure to chitin and driving Th1 type cytokines may be a useful treatment.

Minor:

Figure 1C-D I would recommend using the same y-axis scale if we want to compare AVR vs BTAVR.  

Figure 2D. I would also ask to address the # of donors/samples ran for Fig2A-C. Could we ask for alternative color for TAK242 vs TAK242+LPS it's too similar and hard to distinguish 

Fig 6A. AVR-48 100uM, picture quality is rather low compared to other gated items

B-D, could we request to have the x axis line up?

Did the authors consider other PRRs members that were considered in the recognition of AVR-48  besides CD206 for CLRs ?

I am not sure if the last supplementary figure is needed as it's difficult to note differences as the blood is distracting and matches with the surrounding tissue. 

Round 2

Reviewer 1 Report

The concerns raised in the original review of the manuscript required considerable rewriting of the manuscript or carrying additional experiments. In my opinion, the critique points were not adequately addressed in the revised version. 

Reviewer 2 Report

I'd like to thank the reviewers for addressing most of our concerns brought by myself and the other author. However, I do not feel that the authors understand that we  are very concerned in result figures, particularly as sample size represented doesn't correlate with what's described in the text. 

Example: fig 1C-D. the authors state they perform this study 3 times with 4 technical replicates ( should equal to 12 dots total or at least state it's representative of 1 out of 3 experiments, did the authors show the average of each experiment and that's what they are showing?) both reviewers noted there's only 3-4 dots being represented. This REALLY is concerning, as this is done in most of their figures. 

Example Fig 2B - shows 2 dots for TLR4 concentrations of supernatant but it describes 3 replicates, conducted twice. Again, is this showing the average of the experiment or a representative of 1 of 2 studies? 2 samples should not show statistical significance. The authors should show each replicate for all the figures so that it doesn't raise concerns. 

Fig 3 B authors describe-3 mice, 2 experiments... only shows 3 samples, again is this representative of 1 of 2 experiments? why not show all points and combined experiment results or are they THAT far off, did the inoculum vary? this just raises more questions

Fig 4D - # of macrophages shows 3 dots, but they mentioned 3-5 mice per experiment. Again, show the samples.

Fig8 B,D,F....2-3 representative dots shown...but doesn't represent what is described. The authors need to correct this before it can be published